# Study on Structural Evolution, Thermochemistry and Electron Affinity of Neutral, Mono- and Di-Anionic Zirconium-Doped Silicon Clusters ZrSi*_n_*^0/-/2-^ (*n* = 6–16)

**DOI:** 10.3390/ijms20122933

**Published:** 2019-06-15

**Authors:** Caixia Dong, Limin Han, Jucai Yang, Lin Cheng

**Affiliations:** 1Inner Mongolia Key Laboratory of Theoretical and Computational Chemistry Simulation, School of Chemical Engineering, Inner Mongolia University of Technology, Hohhot 010051, China; Dongcx201011@163.com (C.D.); hanlimin@imut.edu.cn (L.H.); 2School of Mining and Technology, Inner Mongolia University of Technology, Hohhot 010051, China; 3School of Energy and Power Engineering, Inner Mongolia University of Technology, Hohhot 010051, China

**Keywords:** the most stable neutral and Zintl anionic Zr-doped silicon clusters, structural evolution patterns, simulated photoelectron spectroscopy, stability, HOMO-LUMO gap

## Abstract

We have carried out a global search of systematic isomers for the lowest energy of neutral and Zintl anionic Zr-doped Si clusters ZrSi*_n_*^0/-/2-^ (*n* = 6–16) by employing the ABCluster global search method combined with the mPW2PLYP double-hybrid density functional. In terms of the evaluated energies, adiabatic electron affinities, vertical detachment energies, and agreement between simulated and experimental photoelectron spectroscopy, the true global minimal structures are confirmed. The results reveal that structural evolution patterns for neutral ZrSi*_n_* clusters prefer the attaching type (*n* = 6–9) to the half-cage motif (*n* = 10–13), and finally to a Zr-encapsulated configuration with a Zr atom centered in a Si cage (*n* = 14–16). For Zintl mono- and di-anionic ZrSi_n_^-/2-^, their growth patterns adopt the attaching configuration (*n* = 6–11) to encapsulated shape (*n* = 12–16). The further analyses of stability and chemical bonding make it known that two extra electrons not only perfect the structure of ZrSi_15_ but also improve its chemical and thermodynamic stability, making it the most suitable building block for novel multi-functional nanomaterials.

## 1. Introduction

Transition metal (TM) silicides as novel functional materials have been extensively utilized in diverse fields such as aerospace, microelectronic device manufacturing, optical instruments, magnetic material, and the chemical industry because of their oxidation resistance, high-temperature stability, and low electrical resistivity [1,2]. For example, in the early 1960s platinum silicide was successfully used to improve the rectifying characteristics of diodes [3]. Nickel silicide showed significantly enhanced catalytic activity in CO methanation [4]. Composite coatings including molybdenum silicide can be used for protection of missile nozzles and aero engine blades [5]. Zirconium silicide is an ideal device with significant potential applications as anode materials in Li-ion batteries, protective coating on zirconium-alloy fuel cladding in light water reactors, heating elements for electrical furnaces, and also can be incorporated into the semiconductor industry as large scale integrated circuits, Schottky barriers, Ohmic contacts, and rectifying contacts due to the fact that interfaces of zirconium and silicon have good lattice matching, sharp interfaces, low Schottky barrier heights, high conductivity, and excellent thermal stability [1,2,6,7,8]. Therefore, a large amount of research on zirconium silicides was carried out in the past few decades. These investigations concentrate primarily on the formation of amorphous zirconium silicides and have suggested that nanoscales such as nanoclusters, nanotubes, and ultrathin films play a pivotal role in such physical and chemical processes. On the other hand, the miniaturization of devices approaching the sub-nanometer regime could result in material with tunable electronic properties by altering size, shape, and composition based on the application, but it would require a detailed understanding of the interplay between individual transition metals and silicon atoms. Accordingly, it is very significant to explore the evolution of zirconium silicide nanoclusters in the transition from the molecular to the condensed phase.

There have been some previous investigations on Zr-doped silicon clusters. Stimulated by the first observation of TM@Si*_n_* (TM = Cr, Mo, W, *n* = 15 and 16) compounds formed by the laser vaporization supersonic jet expansion technique, the characters of the middle-sized Zr-encapsulated Si clusters were executed theoretically [9]. Kumar et al. calculated the structural and electronic properties of Zr@Si*_n_* (*n* = 8–20) by using the ab initio pseudopotential plane wave method and density functional theory (DFT) with B3PW91 and PW91PW91 schemes and concluded that the evolution pattern of the ground state structure was basketlike open structures for *n* = 8–12, while for *n* = 13–16, it was Zr-encapsulated Si clusters; and for larger-size *n* = 17–20, it was capped Zr@Si_16_ cages [10,11,12,13,14,15]. Sen and Mitas reported on a theoretical study of the electronic structure of clusters with an encapsulated TM atom (TM atom from 3d, 4d, and 5d series) in a Si_12_ hexagonal prism (HP) cage [16]. Lu and Nagase investigated structural and electronic properties of TMSi*_n_* (TM = W, Zr, Os, Pt, Co, etc.) in a range of 8 ≤ *n* ≤ 20 by using B3LYP with LanL2DZ basis sets for the TM atom and the LanL2DZ(d) basis set for Si atoms, and found that the ground state structure of Zr@Si_14_ was a distorted HP with Si_2_ decorating the lateral prism faces, and it was 0.16 eV lower in energy than the distorted HP with two Si atoms symmetrically capping the lateral prism faces [17]. Subsequently, Han and co-workers investigated systematically the total energies, equilibrium geometries, growth pattern mechanisms, and natural population analysis (NPA) of ZrSi*_n_* (*n* = 1–16) by using B3LYP with LanL2DZ basis sets, and proposed that when *n* ≤ 7, the ground state ZrSi*_n_* structures, excluding ZrSi_3_, maintain the same framework as Si*_n_*_+1_, but when *n* = 8–16, the Zr atom evidently disturbs the geometries of the Si clusters, and its localized position gradually changes from the surface site to the concave site of the open Si cage and to the encapsulated site of the sealed Si cage [18]. Wu et al. studied the bond distances, vibrational frequencies, adiabatic electron affinities (AEAs), ionization potential, and dissociation energies of TMSi (TM = 3d, 4d, 5d elements) and their charged ions by using the B3LYP scheme [19]. Furthermore, the geometries and electronic properties of Zr_2_-doped silicon clusters were reported [20,21]. On the experimental aspect, Koyasu et al. investigated the equilibrium configurations, electronic structures, and properties of MSi*_n_* (M = Zr, Sc, Y, Lu, Ti, Hf, V, Nb, and Ta, *n* = 6–20) by means of anion photoelectron spectroscopy (PES) and their reactivity to H_2_O adsorption, presented their AEAs and vertical detachment energies (VDEs), and discussed the cooperative effect between their geometric and electronic structures [22,23]. Recently, bond dissociation energies, valence orbitals, AEAs, and VDEs of ZrSi have been measured by using resonant tow-photon ionization spectra and photoelectron imaging spectroscopy, respectively [1,24,25].

Although these theoretical investigations focused on the structural stability and evolution of Zr-doped neutral Si*_n_* (*n* ≤ 16) clusters and offered significant information for further theoretical and experimental research, a few problems still exist in Zr-doped Si*_n_* species. First, as Maroulis et al. pointed out, the dependence of the predicted ground state structure and property on adopted computational methods and basis sets exists especially for clusters including TM atoms with a partially filled *d* sub-shell [26,27]. The performance of double hybrid (DH) DFT is better than that of pure or single hybrid DFT [28,29]. Considering the sensitivity of basis sets, we choose all-electron basis sets of triple-zeta quality (cc-pVTZ) for Si atoms and ECP28MDF-VDZ (cc-pVDZ-PP) basis sets for Zr atoms. Second, losing the ground state geometry is probable in the period of the initial isomer choice without a global search scheme [30,31,32]. Third, the global minimum is generally verified by comparison of the theoretical and experimental data in detail, owing to the fact that up to now there is no experimental equipment for directly surveying the lowest energy structure of clusters. Fourth, accurate theoretical investigations of neutral and anionic Zr-doped silicon clusters are deficient, especially for the important Zintl anions ZrSi*_n_*^-/2-^. As a matter of fact, only by understanding the ground state structure and the growth pattern of Zr-doped Si clusters can we explain the diversity of their properties. Herein, the unbiased ABCluster global search technique coupled with the mPW2PLYP method and the cc-pVTZ basis set for Si atoms and the cc-pVDZ-PP basis set for Zr atoms is utilized for the configuration optimization of neutral and Zintl anionic ZrSi*_n_*^0/-/2-^ (*n* = 6–16) with the objective of systematically investigating their structural stability and growth patterns, accurately probing the thermochemistry and electron affinity, illuminating details of the bonding characteristic, and offering significant data for further theoretical and experimental investigations of silicon clusters doped with other TM atoms.

## 2. Results and Discussion

### 2.1. Evolution of the Ground State Structure

The global minimal structures and/or competing global minimal isomers optimized with the mPW2PLYP/MIDDLE method for neutral, mono-, and di-anionic ZrSi_n_^0/-/2-^ species are illustrated in Figure 1, Figure 2 and Figure 3, respectively. The corresponding low-lying geometries are illustrated in Appendix A, respectively, where the relative energies from the lowest energy structure calculated at the mPW2PLYP/LARGE//mPW2PLYP/MIDDLE level and point group are also given. Conformational population (%) for low-lying geometries of ZrSi*_n_*^0/-/2-^ species are listed in Appendix A. The average binding energy (*E_b_*), electronic state, and NPA charge on Zr atoms for the global minimum are provided in Table 1.

The ground state of all neutral ZrSi*_n_* (*n* = 6–16) is predicted to be a singlet. The growth pattern adopts the attaching type to the half-cage motif, and finally to the Zr-encapsulated configuration with the Zr atom centered in a Si cage. The global minimum is a face-capped tetragonal bipyramid for ZrSi_6_. For *n* = 7–9, their ground state structure can be viewed as attaching one, two, and three Si atoms to the face of the ZrSi_6_^-^ cluster, respectively. For *n* = 10–13, they are half-cage motifs consisting of two five-membered rings, a seven-membered and four-membered ring, a seven-membered and five-membered ring, and a seven-membered and six-membered ring, respectively. For *n* = 14–16, their ground state structures are encapsulated configurations with the Zr atom located in the Si cage. The cage of ZrSi_14_ is a distorted HP with two Si atoms symmetrically capping the lateral prism faces. The cage comprised of two pentagons and ten quadrilaterals (TPTQ) is predicted to be the global minimum for ZrSi_15_. A fullerene cage is calculated to be the ground state structure for ZrSi_16_. It is noted that the ground state structures for *n* = 7–15 presented herein are different from those reported in [18]. In addition, the most stable structures for *n* = 8–13 differ from those presented in [14], and for *n* = 14 differs from those presented in [15,17].

The ground state for all mono-anionic ZrSi*_n_*^-^ (*n* = 6–16) is calculated to be a doublet. Their structural growth pattern prefers the attaching type to the encapsulated shape with the Zr atom centered in the Si cage. The global minimum of ZrSi_6_^-^ is a distorted pentagonal bipyramid with the Zr atom on the principal axis. For *n* = 7–11, their ground state structures can be regarded as attaching one, two, three, four, and five Si atoms to the face of the ZrSi_6_^-^ cluster, respectively. That is, the distorted pentagonal bipyramid of ZrSi_6_^-^ is the basic structural unit for *n* = 6–11. For *n* = 12–16, their global minima are encapsulated shapes with the Zr atom residing in the Si cage. The HP, capped HP, THSQ (three hexagon and six quadrangle), and fullerene cages are evaluated to be the most stable structures of ZrSi_12_^-^, ZrSi_13_^-^, ZrSi_14_^-^, and ZrSi_16_^-^, respectively. For ZrSi_15_^-^, the TPTQ (15m1) and five-capped PH (pentagonal prism) (15m2) are competitive for the global minimum due to the fact that they are degenerate in energy (the energy difference is only 0.01 eV). Although it is difficult to determine the ground state structure by energy in this situation, the 15m2 isomer is the most probable ground state structure based on simulated PES (see below). To our knowledge, only ZrSi_16_^-^ was studied previously by Wu et al. [33].

The ground state structures for all di-anionic ZrSi*_n_*^2-^ (*n* = 6–16) are evaluated to be singlet. The growth model of di-anionic ZrSi*_n_*^2-^ species is consistent with that of mono-anionic clusters. For *n* = 6–13, 15, 16, the global minima of di-anionic ZrSi*_n_*^2-^ are similar to those of their mono-anions. The cage of di-capped OHFP (one hexagon and four pentagon) is calculated to be the ground state structure for ZrSi_14_^2-^, which differs from that of the corresponding mono-anion.

The distribution of ZrSi*_n_*^0/-/2-^ clusters at room-temperature by Boltzmann statistics can further support our results. From Appendix A, we can see that the lowest-energy structures account for the largest proportion. This verifies the accuracy of the above ground state structures.

### 2.2. PES of Mono-Anionic Species

PES is one of the most important experimental tools to get insight into, and to extract electronic fingerprints from, a variety of atomic and molecular clusters as well as condensed-matter systems. Accordingly, the validity of the predicted ground state structures can be tested by way of comparing their theoretical and experimental PES spectra, in which two norms are used. One is the amount of distinct peaks and their position in the PES spectra, and another is the first VDE and/or AEA. The simulated PES of the global minimal structures coupled with the experimental PES spectra are pictured in Figure 4. The theoretical predicted AEAs and the first VDEs are provided in Table 2 along with experimental data. It can be seen from the simulated PES of ZrSi_6_^-^ in the range of ≤5.8 eV that there are four distinct peaks (X, A–C) residing at 3.32, 3.63, 4.33, and 5.37 eV, of which the first three peaks are in excellent agreement with experimental data of 3.40, 3.60, and 4.30 eV [23]. The simulated PES of ZrSi_7_^-^ has five different peaks (X, A–D) located at 2.86, 3.49, 4.16, 4.85, and 5.59 eV, of which the first three peaks are in excellent accord with the experimental values of 2.80, 3.20, and 4.00 eV [23]. For ZrSi_12_^-^ and ZrSi_16_^-^, there are four discrete peaks (X, A–C) centered at 3.99, 4.30, 5.07, and 5.52 eV, and 2.65, 4.11, 4.70, and 5.61 eV, which reproduce well the experimental data of 4.10, 4.40, 4.90, and 5.50 eV, and 2.82, 4.30, 4.80, and 5.50 eV, respectively [22,23]. For ZrSi_10_^-^, four distinct peaks (X, A–C) residing at 3.35, 3.88, 4.45, and 5.58 eV are obtained. In addition to the first peak (X), the remaining three peaks (A–C) are very close to experimental data of 3.80, 4.90, and 5.60 eV, respectively [23]. For ZrSi_14_^-^, four peaks (X, A–C) centered at 3.18, 3.64, 4.59, and 5.16 eV are also obtained, but only later two peaks appear in the experimental PES and agree well with experimental data of 4.50 and 5.20 eV [23]. For *n* = 8, 9, 11, and 13, their simulated PES has many different peaks. The first 6–8 peaks reside at 2.81, 3.25, 3.84, 4.17, 4.59, and 5.39 eV, 2.76, 3.38, 3.83, 4.43, 4.72, 5.11, and 5.60 eV, 3.25, 3.43, 3.83, 4.27, 4.79, 5.02, 5.39, and 5.66 eV, and 3.53, 3.86, 4.43, 4.82, 5.09, and 5.76 eV, respectively. Unfortunately, only three peaks are observed in their experimental PES. And it seems experimentally that their first peaks are not observed analogous to ZrSi_10_^-^ and ZrSi_14_^-^. For *n* = 15, the simulated PES of 15m1 and 15m2 isomers has four peaks (X, A–C) located at 3.00, 3.93, 4.40, and 4.96 eV, and 3.97, 4.51, 4.94, and 5.62 eV, respectively. Apart from first peak (X), the rest of peaks (A–C) for 15m2 are close to the experimental data of 4.70, 5.10, and 5.50 eV, respectively [23]. On the other hand, the three peaks of the experimental spectra of ZrSi_15_^-^ are also simulated by the calculations with a shift of the first three peaks (X, A, and B) of the 15m2 isomer by 0.7 eV to the higher binding energy [23]. So we suggest that the 15m2 geometries are the most probable ground state structure. Comparing their theoretical and experimental AEAs (see Table 2), we can see that the AEAs of ZrSi*_n_*^-^ (*n* = 6–16) excluding ZrSi_11_^-^, ZrSi_12_^-^, and ZrSi_15_^-^ are agreement with those of experimental data. The mean absolute deviations from experimental values are 0.08 eV, and the largest errors are those of ZrSi_7_^-^ and ZrSi_13_^-^ which are off by 0.11 eV. In short, good agreement between the theoretical and experimental PES spectra sheds further light on the validity of the predicted most stable structures. In light of our reliable theoretical predictions, we suggest that the experimental PES spectra of ZrSi_11_^-^ and ZrSi_15_^-^ species should be checked further. The cause lies in that their theoretical and experimental AEAs and VDEs have large differences.

### 2.3. Stability

The thermodynamic and chemical stability of the global minimal structures of ZrSi*_n_*^0/-/2-^ (*n* = 6–16) species are characterized by the *E_b_*, second energy difference (∆^2^*E*) and the HOMO-LUMO (Highest Occupied Molecular Orbital - Lowest Unoccupied Molecular Orbital) energy gap (*E_gap_*). The *E_b_* and ∆^2^*E* are calculated via the following expressions, where *E* is the total energy of the corresponding atom or cluster. The results are pictured in segments a and b of Figure 5, respectively.
(1)$$E_{b}\left({\mathrm{ZrSi}}_{n}\right)=\left[E\left(\mathrm{Zr}\right)+\mathrm{nE}\left(\mathrm{Si}\right)-E\left({\mathrm{ZrSi}}_{n}\right)\right]/\left(n+1\right)$$
(2)$$E_{b}\left({\mathrm{ZrSi}}_{n}^{-/2-}\right)=\left[E\left(\mathrm{Zr}\right)+\left(n-1\right)E\left(\mathrm{Si}\right)+E\left({\mathrm{Si}}^{-/2-}\right)-E\left({\mathrm{ZrSi}}_{n}^{-/2-}\right)\right]/\left(n+1\right)$$
(3)$$\Delta^{2}E\left({\mathrm{ZrSi}}_{n}^{0/-/2-}\right)=E\left({\mathrm{ZrSi}}_{n-1}^{0/-/2-}\right)+E\left({\mathrm{ZrSi}}_{n+1}^{0/-/2-}\right)-2E\left({\mathrm{ZrSi}}_{n}^{0/-/2-}\right)$$

From Figure 5a we can see the data for *E_b_*: di-anionic compounds > mono-anionic species > neutral clusters. The higher *E_b_*, the more stable the relative stability. It is to say that Zintl anionic Zr-doped Si clusters are more stable than their neutral counterparts, particularly for di-anionic compounds. The cause lies in that neutral ZrSi*_n_* compounds still have a dangling bond. When they obtain electrons, these extra electrons decrease the dangling bonds and improve the stability; (Figure 5b) for neutral ZrSi*_n_* with *n* = 9, 12, and 14, for mono-anionic ZrSi*_n_*^-^ with *n* = 8, 11, and 13, and for Zintl di-anionic ZrSi*_n_*^2-^ with *n* = 7, 10, 12 and 15, the compounds are more stable than their neighbors. These results are obviously reproduced in Figure 5b because the ∆^2^E is a sensitive estimate for relative stability.

*E_gap_* as a significant physical parameter can be regarded as an index of chemical reactivity, particularly for photochemical reactivity. Larger values of the *E_gap_* stand for weaker chemical reactivity. Baerends et al. found that the *E_gap_* predicted by pure DFT is closer to the real optical gap than that calculated by hybrid DFT [34]. In light of this finding, the PBE *E_gap_* of ZrSi*_n_*^0/-/2-^ (*n* = 6–16) compounds are pictured in Figure 5c as functions of the size of Si clusters, and mPW2PLYP *E_gap_* is pictured in Appendix A. It can be seen from Appendix A that the *E_gap_* of mPW2PLYP is larger than that of PBE by 2.46 eV. The *E_gap_* of PBE is a very good approximation to the optical gap. The PBE *E_ga_*_p_ of 1.56 eV for ZrSi_16_ is in good agreement with an approximate experimental value of 1.36 eV [22]. For *n* = 6–9, the di-anionic *E_gap_* is very small and ranges from 0.27 to 0.81 eV, which is narrower than that of corresponding neutral and mono-anionic compounds. The di- and mono-anionic *E_ga_*_p_ for *n* = 10–13, 15, and 16 differ little from each other. For *n* = 14, the mono-anionic *E_ga_*_p_ is smaller than that of the di-anion. The neutral *E_gap_* for *n* = 10 and 14 is very close to that of corresponding di-anion. The neutral *E_gap_* for *n* = 11 and 16 is larger than that of di-anion. The neutral *E_gap_* for *n* = 12, 13, and 15 is smaller than that of di-anion. The *E_gap_* of 2.25 eV for Zintl di-anionic ZrSi_15_^2-^ is the largest among all these compounds. That is, the Zintl di-anionic ZrSi_15_^2-^ cluster possesses not only thermodynamic stability, but also chemical stability, which may make it the most suitable building block for novel multi-functional nanomaterials.

### 2.4. Chemical Bonding Analysis

In order to garner an in-depth comprehension of the ideal chemical and thermodynamic stability of the Zintl di-anionic ZrSi_15_^2-^ five-capped pentagonal prism, the character of the bonding between the Zr atom and the Si_15_^2-^ cage is examined by the AdNDP scheme, which is a generalized NBO (Natural Bond Orbital) search strategy to analyze the delocalized and localized multicenter bonds (encoded as nc-2e, where n can range from 1 (lone pair) to the number of atoms in the cluster) introduced by Zubarev and Boldyrev [35]. As can be seen from Figure 6, the chemical bonding of 66 valence electrons can be split into five types: lone-pair, 2c-2e, 3c-2e, 4c-2e, 10c-2e, and 16c-2e. Each capped Si atom has a lone pair. The pentagonal prism is characterized by ten 2c-2e localized Si-Si σ bonds with 1.86 |e| in each bond. The ten 3c-2e, five 4c-2e, two 10c-2e, and 16c-2e delocalized π+σ bonds with 1.79–2.00 |e| in each bonds are accountable for the interplay between the inner Zr atom and the outer shell of Si_15_ and enhance the stability of the encapsulated ZrSi_15_^2-^ molecule.

## 3. Conclusions

The systematic isomer search for low energy structures of neutral, mono-, and di-anionic Zr-doped Si clusters ZrSi*_n_*^0/-2-^ (*n* = 6–16) are executed by employing an ABCluster global search technique combined with a mPW2PLYP double hybrid density functional scheme. In terms of the evaluated energies, AEA, VDE, and agreement between simulated and experimental PES, the true global minimal structures are confirmed. The results reveal that structural evolution patterns for neutral ZrSi*_n_* clusters prefer the attaching type (*n* = 6–9) to the half-cage motif (*n* = 10–13), and finally to the Zr-encapsulated configuration with the Zr atom centered in a Si cage (*n* = 14–16). For Zintl mono- and di-anionic ZrSi*_n_*^-/2-^, their growth patterns adopt the attaching configuration (*n* = 6–11) to the encapsulated shape (*n* = 12–16). The calculated AEA for ZrSi*_n_*^-^ (*n* = 6–16) with the exception of *n* = 11, 12, and 15 are in excellent accord with those of experimental data. The mean absolute deviations from experimental values are 0.08 eV. The further analyses of stability and chemical bonding make it known that two extra electrons not only perfect the structure of ZrSi_15_ but also improve its chemical and thermodynamic stability, which may make it the most suitable building block for novel multi-functional nanomaterials. We think that this study will provide strong motivation for further verification of experimental photoelectron spectroscopy of ZrSi_11_^-^ and ZrSi_15_^-^ and for further experimental and theoretical studies of other transition metal-doped Si clusters.

## 4. Theoretical Methods

We utilized three schemes to search the initial isomers of neutral and Zintl anionic ZrSi*_n_*^0/-/2-^ (*n* = 6–16) in order to search for their true global minimum. First, the calculations started from unbiased searches for the low-lying structures of series ZrSi*_n_*^0/-^ (*n* = 6–16) (with the exception of ZrSi*_n_*^2-^, of which isomers were chosen in light of ZrSi*_n_*^-^) through the ABCluster global search scheme coupled with the GAUSSIAN 09 software suite [36,37]. At least 300 configurations produced by the ABCluster algorithm for each ZrSi*_n_*^0/-^ (*n* = 6–16) species were optimized at the PBE [38] level with a cc-pVDZ-PP basis set for Zr atoms and a 6–31G basis set for Si atoms [39]. Second, the “substitutional structure”, which is derived from the ground state structure of Si*_n_*_+1_ by replacing a Si atom with a Zr atom, was utilized. Third, the structures reported in the preceding publications were utilized [9,10,11,12,13,14,15,16,17,18]. Then, the selected low-lying structural candidates were reoptimized at the PBE level with the MIDDLE (cc-pVTZ basis set for Si atoms and basis set for Zr atoms unchanged) basis sets. Vibrational frequency evaluations were also executed at the same level to make sure all the optimized isomers were true local minimum structures. After accomplishment of the preliminary structure optimization through the PBE scheme, the low-lying structural candidates were, again, selected and reoptimized by using the DH-DFT of mPW2PLYP with the MIDDLE basis set [40]. The mPW2PLYP frequency was not calculated. To further refine the energies, single-point energy calculations were finally executed at the mPW2PLYP level with the LARGE (aug-cc-pVTZ basis set for Si atoms and basis set for Zr atoms unchanged) basis set. Moreover, NPA were also executed to further comprehend the interplay between the Zr atom and Si clusters. The PES spectra of the negatively charged ions ZrSi*_n_*^-^ (6–16) were simulated in terms of Koopman’s theorem at the mPW2PLYP/LARGE level via the Multiwfn software suite and compared with the experimental ones [41,42]. The spin multiplicity of singlets, triplets, and quintuplets were considered for neutral ZrSi*_n_* and di-anionic ZrSi*_n_*^2-^ (*n* = 6–16), and doublets and quartets were considered for negatively charged ZrSi*_n_*^-^ (*n* = 6–16) clusters. To check the reliability of the calculations, the ZrSi structures and properties predicted by mPW2PLYP and CCSD(T) were compared and provided in Table 3. From Table 3 we can see that the order of increasing energy predicted by mPW2PLYP is singlet, to quintet, to the triplet state, which is in agreement with the result evaluated by CCSD(T), while B3LYP is not. On the other hand, the bond dissociation energy (D_0_), AEA, and VDE calculated by the mPW2PLYP method were 2.74, 1.44, and 1.45 eV, respectively, which are in good accordance with experimental values of 2.95 [25], 1.46 [24], and 1.58 eV [24]. In particular, the D_0_ and AEA predicted by mPW2PLYP among these methods listed in Table 3 is the closest to the experimental data. As a matter of fact, selection of calculation method and basis set is hardly a trivial matter. The cause lies in that the structures and properties predicted by different methods and basis sets may be different. There are many such example. For instance, at the CCSD(T) level, the LanL2DZ basis set for Si atom predicted the ground state of ZrSi was to be triplet state, but 6-311+G(3df,3pd) basis set for Si atom predicted to be singlet state [1].

## Figures and Tables

**Figure 1 ijms-20-02933-f001:**
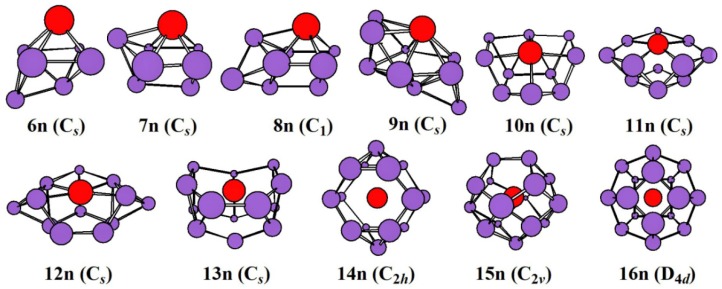
The lowest-energy structures of ZrSi*_n_* (*n* = 6–16) and their point group.

**Figure 2 ijms-20-02933-f002:**
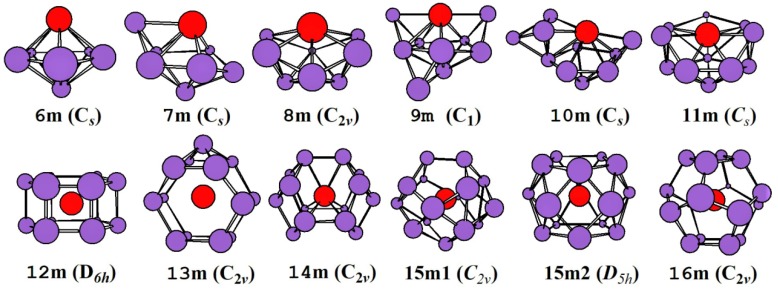
The lowest-energy structures of mono-anionic ZrSi*_n_*^-^ (*n* = 6–16) and their point group.

**Figure 3 ijms-20-02933-f003:**
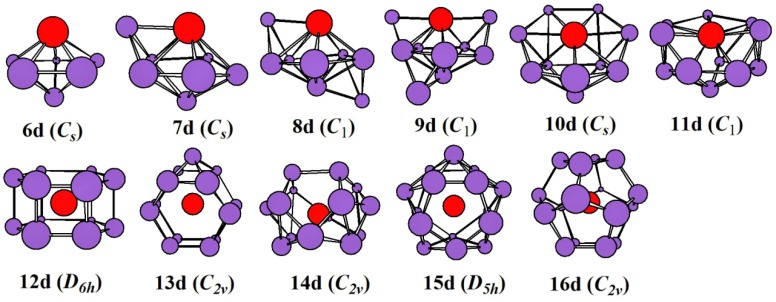
The lowest-energy structures of di-anionic ZrSi*_n_*
^2-^ (*n* = 6–16) and their point group.

**Figure 4 ijms-20-02933-f004:**
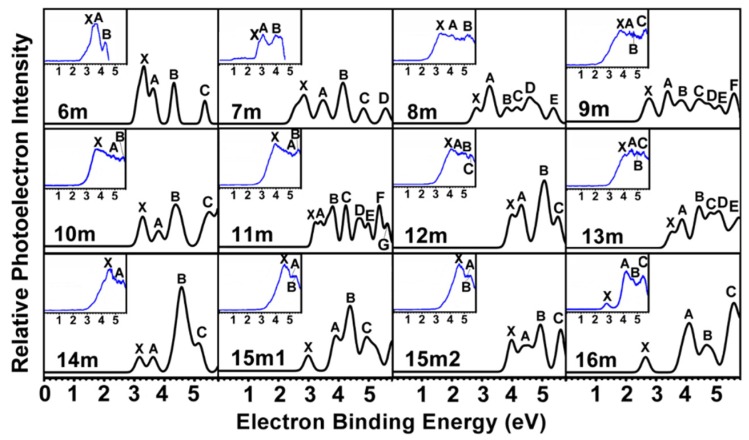
Photoelectron spectroscopy (PES) spectra of the most stable ZrSi*_n_*^-^ (*n* = 6–16) clusters. The inserts show the experimental PES spectra taken from [22,23].

**Figure 5 ijms-20-02933-f005:**
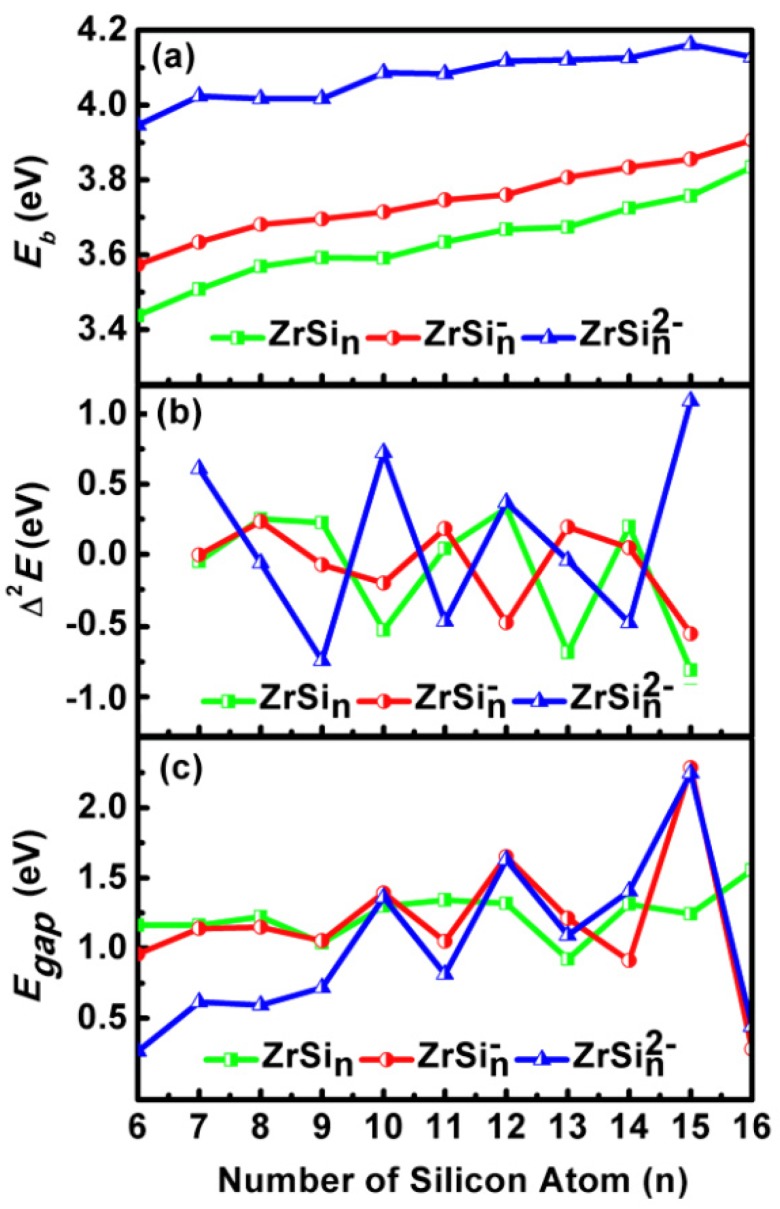
Size dependences of (**a**) average bonding energy (*E_b_*). (**b**) second energy difference (∆^2^*E*), and (**c**) HOMO-LUMO energy gap (*E_gap_*) for the ground state ZrSi*_n_*^0/-/2-^ (*n* = 6–16) clusters.

**Figure 6 ijms-20-02933-f006:**
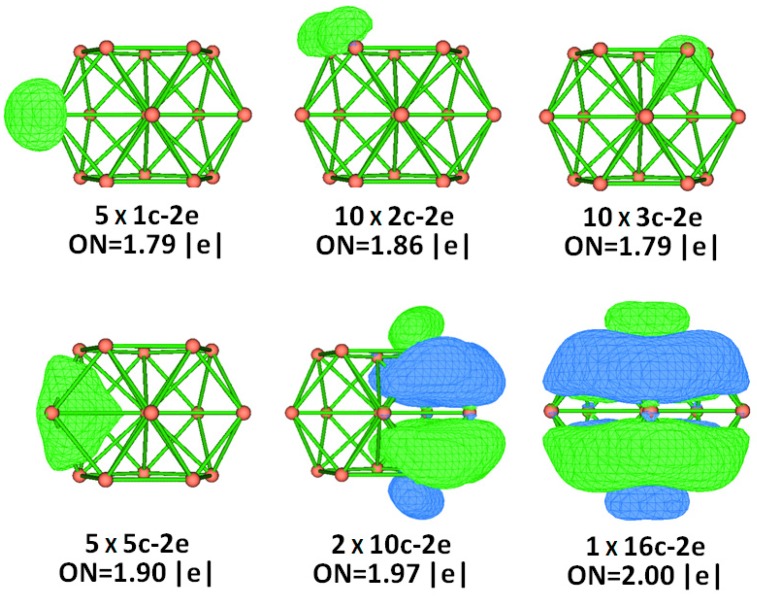
AdNDP analysis of ZrSi_15_^2-^ cluster. ON stands for the occupation number.

**Table 1 ijms-20-02933-t001:** Electronic state, average bonding energy E_b_ (eV), and the charge on the Zr atom Q(Zr) (a.u.) of the ground state structure of ZrSi*_n_*^0/-/2-^ (*n* = 6–16) clusters.

	ZrSi*_n_*			ZrSi*_n_*^-^			ZrSi*_n_*^2-^		
*n*	State	Eb	Q(Zr)	State	Eb	Q(Zr)	State	Eb	Q(Zr)
6	^1^A_1_	3.44	−0.13	^2^A’’	3.57	0.25	^1^A′	3.95	−0.49
7	^1^A′	3.51	−0.34	^2^A’	3.63	0.04	^1^A′	4.02	−0.58
8	^1^A	3.57	−0.16	^2^B_2_	3.68	−0.40	^1^A	4.02	−0.31
9	^1^A_1_	3.59	−0.50	^2^A	3.7	−0.20	^2^A	4.02	−0.70
10	^1^A′	3.59	−0.53	^2^A′	3.71	−0.40	^1^A′	4.09	−0.90
11	^1^A′	3.63	−0.98	^2^A’’	3.75	−0.60	^1^A	4.08	−0.76
12	^1^A′	3.67	−0.87	^2^A_2*u*_	3.76	−1.20	^1^A_1*g*_	4.12	−1.33
13	^1^A′	3.67	−1.87	^2^A_1_	3.81	−1.20	^1^A_1_	4.12	−1.24
14	^1^A_g_	3.72	−2.66	^2^B_1_	3.83	−1.20	^1^A_1_	4.13	−1.45
15	^1^A_1_	3.76	−3.31	^2^A_1_′	3.85	−1.29	^1^A_1_′	4.16	−2.90
16	^1^A_1_	3.83	−1.89	^2^B_1_	3.91	−0.90	^1^A_1_	4.13	−0.77

**Table 2 ijms-20-02933-t002:** Theoretical and experimental adiabatic electron affinity (AEA) and vertical detachment energy (VDE) for ZrSi*_n_*^-^ (*n* = 6–16).

	AEA		VDE	
Species	Theor	Exptl	Theor	Exptl
6n←6m	2.24	2.30 ± 0.0043 *^a^*	3.32	3.4 ± 0.1 *^a^*
7n←7m	2.29	2.40 ± 0.0043 *^a^*	2.86	2.8 ± 0.1 *^a^*
8n←8m	2.31	2.30 ± 0.0043 *^a^*	2.81	3.3 ± 0.1 *^a^*
9n←9m	2.29	2.20 ± 0.0043 *^a^*	2.76	3.9 ± 0.1 *^a^*
10n←10m	2.64	2.70 ± 0.0043 *^a^*	3.35	3.8 ± 0.1 *^a^*
11n←11m	2.65	2.90 ± 0.0043 *^a^*	3.25	4.0 ± 0.1 *^a^*
12n←12m	2.50	2.90 ± 0.0043 *^a^*	3.99	4.1 ± 0.1 *^a^*
13n←13m	3.11	3.00 ± 0.0043 *^a^*	3.53	4.1 ± 0.1 *^a^*
14n←14m	2.90	3.00 ± 0.0043 *^a^*	3.18	4.5 ± 0.1 *^a^*
15n←15m1	2.87	3.20 ± 0.0043 *^a^*	3.00	4.7 ± 0.1 *^a^*
15n←15m2	2.86	3.20 ± 0.0043 *^a^*	3.97	4.7 ± 0.1 *^a^*
16n←16m	2.52	2.46 ± 0.080 *^b^*	2.65	2.9 ± 0.1 *^a^*

***^a^*** The data taken from [23]. ***^b^*** The data taken from [22].

**Table 3 ijms-20-02933-t003:** The relative energy of ZrSi between different spin multiplicities along with predicted adiabatic electron affinity (AEA), vertical dissociation energy (VDE), and bond dissociation energy (D_0_) *^a^.*

	Spin Multiplicity			Property		
Method	1	3	5	AEA	VDE	D_0_
mPW2PLYP/LARGE// mPW2PLYP/MIDDLE	0.00	0.26	0.17	1.44	1.45	2.74
CCSD(T)/LARGE// mPW2PLYP/MIDDLE	0.00	0.26	0.18	1.55	1.56	2.64
B3LYP/LANL2DZ *^b^*	0.13	0.13	0.00	1.58	1.61	2.51 *^c^*
B3LYP/mixed basis Set *^b^*	0.12	0.08	0.00	1.55	1.59	
CCSD(T)/mixed basis set *^b^*	0.00	0.13	0.10			
Exp.				1.46 *^d^*	1.58 *^d^*	2.95 *^c^*

***^a^*** The relative energy, AEA, VDE, and D_0_ is in eV. The AEA and VDE were calculated based on the ^2^∑^+^ ground state of anionic ZrSi^-^; ***^b^*** The data taken from [1]; ***^c^*** The data taken from [25]; ***^d^*** The data taken from [24].

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
