# Peer review of "Study on Structural Evolution, Thermochemistry and Electron Affinity of Neutral, Mono- and Di-Anionic Zirconium-Doped Silicon Clusters ZrSi*_n_*^0/-/2-^ (*n* = 6–16)"

_ijms, 2019, doi:10.3390/ijms20122933_

Round 1

Reviewer 1 Report

This paper by Dong et al reports an extensive theoretical investigation of the molecular structure and properties of Zr-doped medium silicon clsuters. Added to previous findings by other authors on doped slicon clsuters, the present results improve significantly our knowledge on such structures. This paper will be of interest to a large spectrum of researchers active in the field, experimentalists and theoreticians alike.

The authors have done well. This is a work of maturity. Their experience shows clearly here. 

I see little tha would need change or revison.

My only suggestion to the authots is to add a few sentences on the basis set dependence of the calculated values. Admittedly, the properties studied here should sho little basis set sensitivity. Nevertheless, is both useful and instructive to enlighten the readers on this important point

Significant work. Publication recommended after some minor revision, as suggested above.

Author Response

Author reply: We are grateful to the encouraging comments and suggestions from the reviewer. Based on the comments, we added comments (page 2 lines 90,91 and page 3 lines 99,100).

Added sentences: "Considering the sensitivity of basis set, we choose all-electron basis sets of triple-zeta quality (cc-pVTZ) for Si atoms and ECP28MDF-VDZ (cc-pVDZ-PP) basis set for Zr atoms." "Herein, the unbiased ABCluster global search technique coupled with mPW2PLYP method and cc-pVTZ basis set for Si atoms and cc-pVDZ-PP basis set for Zr atoms is utilized for the configuration optimization of neutral and Zintl anionic ZrSin0/-/2- (n=6-16) with the objective of systematical investigating their structural stability and growth pattern, accurate probing the thermochemistry and electron affinity, detailed illuminating the bonding characteristic, and offering significant data for farther theoretical and experimental investigations of silicon clusters doped with other TM atoms."

We hope the above statements answer the reviewer’s comments, and we welcome any further comments. 

Reviewer 2 Report

In this work authors have carried out systematic isomers global search for the low energy of neutral and 16 Zintl anionic Zr-doped Si clusters ZrSin0/-/2- (n=6-16) by employing the ABCluster global search 17 method combined with the mPW2PLYP double-hybrid density functional. The work is interesting and earns probably high interest justifying publication in the International Journal of Molecular Sciences. The references are upto date, the structure of the manuscript is appropriate, the data collected by the techniques applied support the conclusions. I suggest publications after minor revision considering the following remarks:

Authors made conclusion on the stabilities on the base of bonding energies of different clusters, which energies, however differs from each other in a few 10 meV. Considering that the calculations performed at zero Kelvin and considering the kinetic energies associated to each freedom of movements at room-temperature, how far the conclusion can be kept true ? I suggest authors to examine this question and insert few sentences into the manuscript accordingly. I suggest also insert the distribution of clusters occupied at room-temperature by Boltzmann statistics.

Minor remarks:  

Please check the Experimental Adiabatic Electron Affinity (AEA) data in Table 3. They are probably correct, but too many digits are zero.

Author Response

Author reply: Based on the reviewer's comments, we added comments (page 5 lines 193-195) in the revised manuscript and conformational population (%) for low-lying geometries of ZrSin0/-/2- species listed in Table S1 in Supplementary Information.

Added sentences: "The distribution of ZrSin0/-/2- clusters at room-temperature by Boltzmann statistics can further support our results. From Table S1, we can see that the lowest-energies structures account for the largest proportion. This verifies the accuracy of the above ground state structures."

2. Please check the Experimental Adiabatic Electron Affinity (AEA) data in Table 3. They are probably correct, but too many digits are zero.

Author reply: We have checked the AEA data in Table 3 and found that they were correct. The redundant zeros have been deleted.

We hope the above statements answer the reviewer’s comments, and we welcome any further comments. 
